

# How does sea ice influence $\delta^{18}O$ of Arctic precipitation?

Anne-Katrine Faber[1], Bo Møllesøe Vinther [1], Jesper Sjolte[2], and Rasmus Anker Pedersen[1,3]

[1]Centre for Ice and Climate, Niels Bohr Institute, University of Copenhagen, Copenhagen, Denmark
[2]Department of Geology, Quaternary Sciences, Lund University, Lund, Sweden
[3]Climate and Arctic Research, Danish Meteorological Institute, Copenhagen, Denmark

*Correspondence to:* Anne-Katrine Faber (akfaber@nbi.ku.dk)

**Abstract.** This study investigates how variations in Arctic sea ice cover influence $\delta^{18}O$ of present-day Arctic precipitation. This is done using the model isoCAM3, an isotope-equipped version of the National Center for Atmospheric Research Community Atmosphere Model version 3. Four sensitivity experiments and one control simulation are performed with prescribed SST and sea ice. Each of

the four experiments simulates the atmospheric and isotopic response to Arctic oceanic conditions for selected years after the beginning of the satellite era in 1979.

Results show that $\delta^{18}O$ of precipitation is sensitive to local changes of sea ice concentration. Reduced sea ice extent yields more enriched isotope values while increased sea ice extent yields more depleted isotope values. The distribution of the sea ice cover is essential for the spatial distribution

of the simulated changes in $\delta^{18}O$. The experiments of this study show no changes of $\delta^{18}O$ for central Greenland. However, this does not exclude that simulations based on other sea ice distributions might yield changes in Greenland $\delta^{18}O$.

## 1   Introduction

Records of stable water isotopes from polar ice cores have been widely used to reconstruct past

climate variability. Since the pioneering work by Dansgaard (1964), the understanding of stable water isotopes as a proxy for temperature has been significantly advanced. It has become clear that the isotopic composition of precipitation is a complex signal, influenced by both local and regional climate conditions (Vinther et al., 2010; Steen-Larsen et al., 2011). The isotopic composition of the precipitation is integrated along the moisture transport pathway from source to deposition. As

a result, there is a need for a detailed process-based understanding of the factors that can alter the isotopic composition of the transported moisture.

Studies using models, ice cores, snow and vapour measurements have investigated the physical and dynamical processes influencing the isotopic composition of precipitation. Variations in local



Greenland temperatures, conditions at source regions and atmospheric circulation all influence the

isotopic composition of Greenland precipitation (Steen-Larsen et al., 2011; Bonne et al., 2014; Sodemann et al., 2008a, b; Sjolte et al., 2011; Werner, 2010; Vinther et al., 2010).

Several model studies highlight sea ice changes as important for understanding changes in the isotopic composition of precipitation. Sea ice changes in the Arctic were investigated during Dansgaard-Oeschger events (Li et al., 2010) and for exceptionally warm climates (Sime et al., 2013). For Antarc-

tica, the impact of sea ice changes were studied using idealized reductions of the circular shaped sea ice cover (Noone, 2004). None of these model studies investigate sea ice perturbations comparable to observed changes during the present-day climate. Observations from ice cores in this period demonstrate that sea ice changes can influence the isotope composition of precipitation (Divine et al., 2011; Opel et al., 2013; Küttel et al., 2012; Fauria et al., 2010).

A study of idealised changes of Antarctic sea ice show a non-uniform spatial distribution of the modelled isotopic response over Antarctica (Noone, 2004). The heterogeneity of the response is suggested to reflect the existence of different processes driving local and long range moisture transport to coastal and high elevation regions of Antarctica. Due to differences in the configuration of landmasses, open ocean and sea ice, it is difficult to directly transfer findings of Noone (2004) from

Antarctica to the Arctic.

Sea ice changes in the Arctic are found to influence the isotopes in Greenland precipitation. The impact of changes in sea ice and connected sea surface temperatures (SST) of the Arctic ocean were studied by Sime et al. (2013). The sea ice conditions were created using a coupled experiment where a climate model was forced by respectively $2\times$, $4\times$ and $8 \times CO_2$. Hereafter the sea ice and

SST conditions were used to force the applied atmospheric isotope models. Differences in the configurations of sea ice extent and SST were found essential for the resulting large variability in the isotope-temperature slope of $0.1 - 0.7\,‰/C$ for the Greenland ice sheet. While these $CO_2$ changes used by Sime et al. (2013) do not allow direct comparison with present-day Arctic conditions, the results highlights processes that might be important for present day climate.

The recent decades of rapid Arctic sea ice decline provide an interesting opportunity to study how $\delta^{18}O$ respond to realistic sea ice changes of present-day climate. We here present results from isoCAM3 model simulations forced with observed Arctic sea ice and sea surface temperature (SST) conditions derived from observations. This paper will address how the sea ice extent influence the $\delta^{18}O$ in precipitation in the Arctic, and if the response depends on the configuration of the sea ice

changes. Furthermore, the processes responsible for observed changes in $\delta^{18}O$ will be investigated. This is done for several configurations of sea ice to assess the robustness of the response to different magnitudes of sea ice concentration changes and spatial patterns. The structure of the paper is as follows. First the model and experiments are described. Hereafter result of the atmospheric and isotopic response are presented. Then the influence of atmospheric moisture processes is investigated

and discussed.



## 2 Experimental configuration

### 2.1 The model isoCAM3

The simulations of the isotopic composition of precipitation and water vapour in this study are conducted with isoCAM3 (Noone and Simmonds, 2002; Noone, 2003; Noone and Sturm, 2010).
This is an atmospheric general circulation model (AGCM) enabled with the ability to trace the various species of water isotopes. The model is based on the Community Atmosphere Model version 3 (CAM3) (Collins et al., 2006) with a third-generation isotope scheme.

Model-data comparison using the database Global Network of Isotopes in Precipitation (GNIP) show that isoCAM3 simulates the spatial distribution of $\delta^{18}O$ in agreement with observed spatial
isotopic patterns (Noone, 2003). Furthermore, isoCAM3 has been applied in several studies that investigated the isotopic response to past climate changes (Tharammal et al., 2013; Speelman et al., 2010; Sturm et al., 2010; Pausata et al., 2011; Liu et al., 2012b; Sewall and Fricke, 2013; Liu et al., 2014).

The horizontal resolution of the model is T85 ($\sim 1.4°$ x $1.4°$) with 26 hybrid-sigma levels in the
vertical. In this study the SST and sea ice concentrations are specified, thus the only temperatures that are calculated interactively are land and sea ice surface temperatures. This configuration thus allows no feedback between atmospheric circulation and open ocean SST. Greenhouse gases, vegetation, ice sheets are all set to modern values. More specifically greenhouse gasses are set to the following CAM3 default levels (year 1990): $CO_2$: 355 (ppmv), $CH_4$: 1714 (ppbv), $N_2O$: 311 (ppbv). The
solar constant is set to 1365 ($Wm^{-2}$) and orbital configurations are set to the year 1850.

### 2.2 Ensemble design

We perform a set of four sensitivity experiments and one control simulation to investigate how observed variations in Arctic sea ice cover influences $\delta^{18}O$. Every model integration is run for 15 years (following one year for spin-up).Each of the four sensitivity experiments simulates the $\delta^{18}O$
response to sea ice concentration and sea surface temperature (SST) for selected years spanning the time period 1979-2013 within the satellite era. The 12 months time periods are selected based on the four most extreme cases of high and low September sea ice extent recorded during the time period (1979-2012) by the NSIDC Sea ice Index (Fetterer et al., 2002) [updated daily]. The control simulation (CTRL) simulates the $\delta^{18}O$ response using the 12 months climatology of sea ice concentration
and SST for the full time period April 1979 to March 2013. Only the Arctic oceanic surface boundary conditions differ between the runs. An overview of the model experiments are given in table 1.

We force the model isoCAM3 with an annual cycle of monthly mean SST and sea ice conditions obtain from ERA-Interim (Dee et al., 2011) . This annual cycle goes from April to March thus
spanning the full sea ice cycle related to the selected cases of September sea ice extent. Here after the



**Table 1.** Overview of model experiments

| Experiment | Prescribed SST and sea ice |
| --- | --- |
| "1980" | ERA-Interim monthly mean: April 1980-March 1981 |
| "1996" | ERA-Interim monthly mean: April 1996-March 1997 |
| "2007" | ERA-Interim monthly mean: April 2007-March 2008 |
| "2012" | ERA-Interim monthly mean: April 2012-March 2013 |
| CTRL | ERA-Interim monthly mean climatology: April 1979-March 2013 |

model runs for 15 years (following one year of spin up) with repeated annual cycle. The re-analysis data are interpolated bilinearly from the ERA-Interim ( $1°$ x $1°$) to the CAM3 T85 resolution, and hereafter checked for consistency.

Changes in Arctic SST are in nature an inseparable part of the sea ice changes. Keeping the SST constant and only simulating the atmospheric response to sea ice changes, would therefore lead to unrealistic temperature gradients (see Screen et al. (2013b) for further discussion on this topic). Therefore we chose that these experiments are based on both changes in sea ice and SST. A masking of the SST data is applied to eliminate remote influences from extra-polar climate patterns (e.g. from the El Niño Southern Oscillation or Pacific Decadal Oscillation).This masking is constructed so that only the conditions near the Arctic differ from experiment to experiment. Hence this global ocean data is divided in an Arctic and a non-Arctic region. The Arctic region refers to the region of ocean/sea ice conditions expected to influence the Arctic climate and is therefore rather semi-Arctic. Due the geographical configuration of the continents it is chosen to confine this region with southern boundaries of $66°N$ and $37°N$ for the Pacific and Atlantic sector respectively. The relative southern definition of the semi-Arctic region in the North Atlantic is chosen to also include the southern position of sea ice export in the Newfoundland area.

Each experiment is forced by different SST and sea ice conditions in the (semi-)Arctic region corresponding to the values for the selected year. The non-Arctic part of the dataset is identical for all the different experiments and has values from the mean climatology of ERA-Interim 1979-2012. No smoothing is applied between the Arctic and non-Arctic as this would also smooth out naturally occurring SST gradients. The sea ice concentrations and SST used to force the model are shown in Fig. 1 and Fig. 2 here displayed as annual mean anomalies between the respective experiment and the CTRL run.

## 3   Atmospheric response to changes in sea ice extent

### 3.1   Atmospheric response

Changes in sea ice concentration and SST forces a strong local response in surface air temperature ($T_{2m}$) (see Fig. 3) with cooling when sea ice extent is increased and warming when sea ice extent





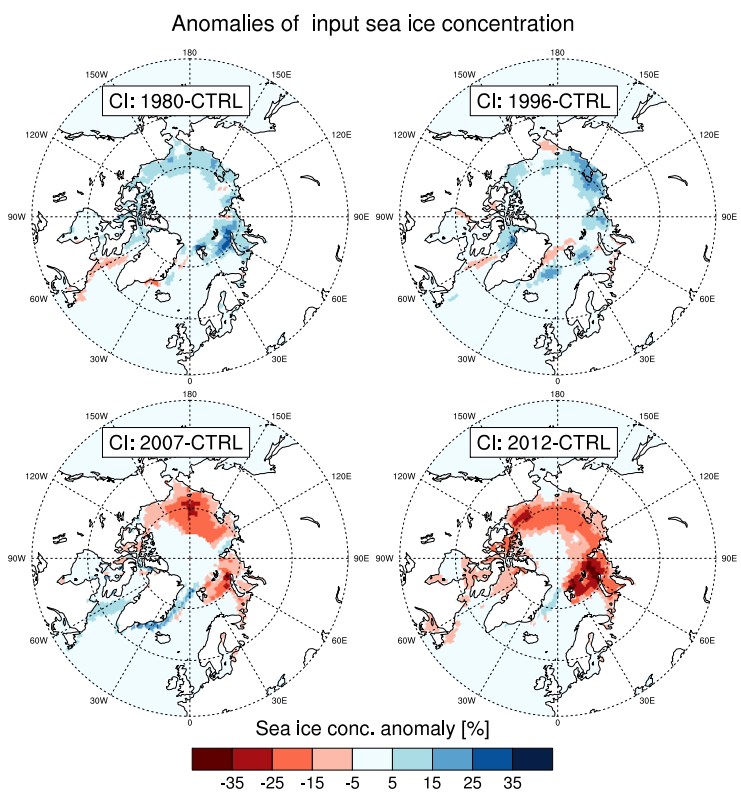

**Figure 1.** Annual mean anomalies of sea ice concentration from ERA-Interim used to force the model. Red colours represent a decrease in sea ice compared to the CTRL run. Blue colours represent an increase in sea ice compared to the CTRL run (mean April 1979 to March 2013).





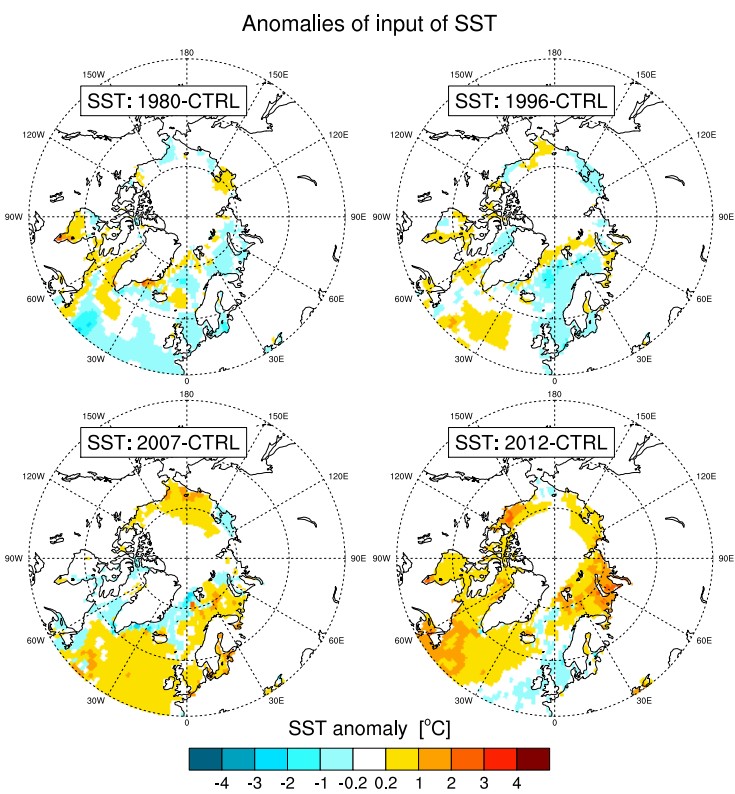

**Figure 2.** Annual mean anomalies of SST data from ERA-Interim used to force the model. Red and yellow colours represent a increase in SST compared to the CTRL run. Blue colours represent a decrease in SST compared to the CTRL run (mean April 1979 to March 2013).



is decreased. The simulated temperature changes are in agreement with other modelling studies that
have investigated the atmospheric response to prescribed reanalysis-based sea ice changes (Screen

et al. (2013a); Magnusdottir et al. (2004); Blüthgen et al. (2012),see also reviews Budikova (2009);
Bader et al. (2011)).

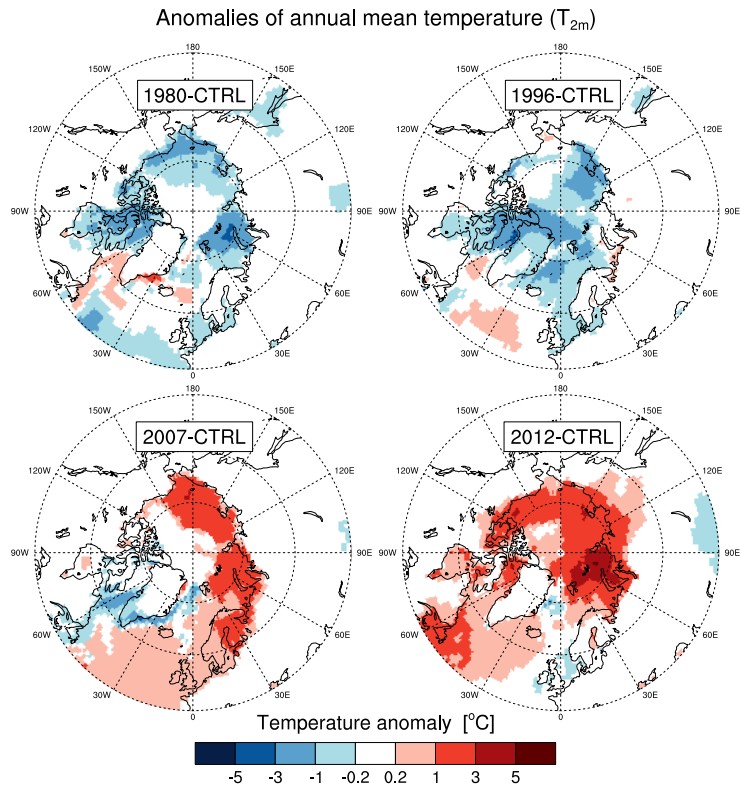

**Figure 3.** Surface air temperatures ($T_{2m}$) for the four simulations compared to the CTRL run. Only anomalies
statistical significant at the 95% confidence level are shown.



### 3.2 Isotopic response

All sensitivity experiments clearly show that changes in sea ice influence the modelled $\delta^{18}O$ of Arctic precipitation (Fig. 4). Decreased (increased) sea ice extent and concentration results in enriched
(depleted) $\delta^{18}O$ values of precipitation (hereafter referred to as $\delta^{18}O_p$). The spatial distribution of changes in $\delta^{18}O_p$ match the spatial distribution of changes in simulated surface air temperature.

The anomalies of $\delta^{18}O_p$ and the SST and sea ice anomalies have similar spatial distributions. This show that the spatial response of the simulated $\delta^{18}O_p$ to sea ice changes is controlled by the distribution of the sea ice changes. The distribution of the $\delta^{18}O_p$ response to the ocean conditions
depends on the sea ice and SST configuration in the different experiments. As shown in Fig. 4 the $\delta^{18}O_p$ of the precipitation over central part of Greenland appears unaffected by the simulated changes in sea ice cover in all experiments whereas $\delta^{18}O_p$ changes over the Pacific-Arctic and the Barents/Kara Sea region depend on the distribution of sea ice in the given experiment.

The experiments "1980" and "1996" both have increased sea ice extent compared to the CTRL
experiments, yet the spatial distribution of the sea ice changes in the Arctic Ocean are very different. This is observed in the Barents/Kara Sea region, in the Baffin Bay and near the northern coast of Greenland. The corresponding isotopic response match the differences in spatial pattern observed in sea ice cover.

The two experiments with low sea ice extent compared to the CTRL experiments (the "2007" and
"2012" experiments) similarly show that sea ice distribution is important for $\delta^{18}O_p$. The Labrador/Baffin region does not experience any significant change in the isotopic composition of precipitation in the "2007" experiment but it does in the "2012" experiment where the sea ice changes in this region are much more pronounced. For the Barents Sea regions both experiments yield positive $\delta^{18}O_p$ anomalies, but the amplitude of the anomalies is different. Interestingly this difference in amplitudes is
also found in the sea ice concentration anomalies used to simulated the isotopic response. Thus this suggests that both distribution and magnitude of the sea ice changes are important for the sea ice controlled change in $\delta^{18}O_p$.

### 3.3 $\delta^{18}O_p$-temperature relationship

From a climate reconstruction perspective it is interesting to examine whether the $\delta^{18}O_p$-T slope is
sensitive to changes in sea ice cover and SST. Scatter plots of annual mean anomalies of $\Delta\delta^{18}O_p$-$\Delta T$ are shown in Fig. 5. The same plot with precipitation-weighted $\delta^{18}O_{p.wgt.}$ is shown in Fig. 6. Only grid points in the Arctic ( $60°N$ - $90°N$ ) are included in the analysis.

Linear regression shows that the spatial $\Delta\delta^{18}O_p$-$\Delta T$ slope for each of the experiment all are within the range of 0.38 to 0.53 ‰/$°C$ for all experiments. Linear regression for all experiments
together (Fig. 5 "All experiments" ) show a larger range of values of anomalies and yields a slope of 0.57 ‰/$°C$ with $R^2 = 0.761$.





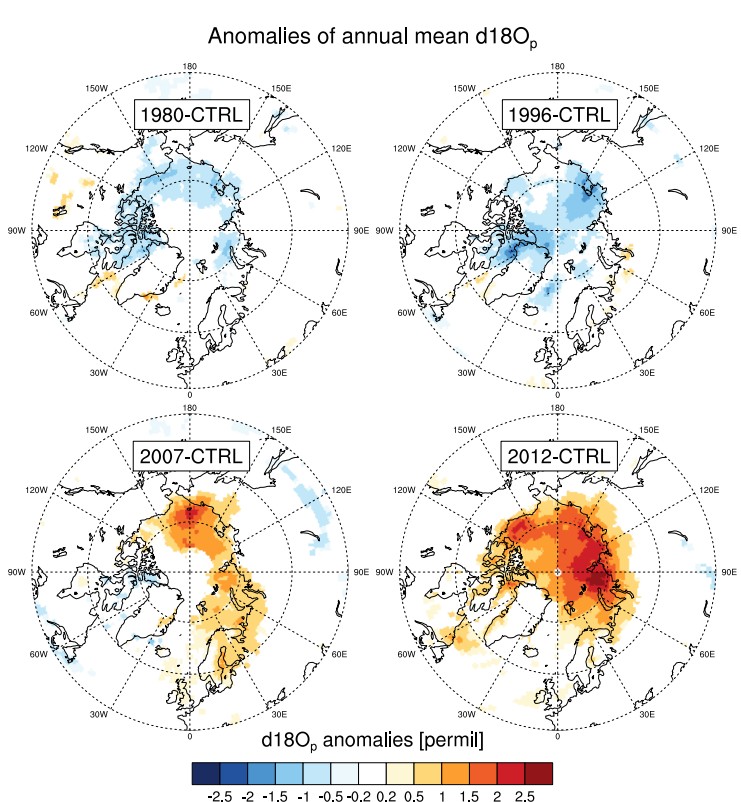

**Figure 4.** $\delta^{18}O$ of precipitation ($\delta^{18}O_P$) for the four simulations compared to the CTRL run. Only anomalies statistical significant at the 95% confidence level are shown.





For experiments with high sea ice extent the slope is 0.38 ‰/$°C$ with $R^2 = 0.59$ for "1980" and 0.53 ‰/$°C$ with $R^2 = 0.575$ for "1996". The cases with low sea ice extent have values of the slope, 0.42 ‰/$°C$ with $R^2 = 0.732$ for "2007" and 0.48 ‰/$°C$ with $R^2 = 0.635$ for "2012".

Results of linear regression using $\delta^{18}O_{p.wgt}$ yields values of $R^2$ in the range of 0.3 - 0.6 for the individual experiments. The best fit for the regression model is found for regression with all experiments together, as the span of changes in temperature and $\delta^{18}O_{p.wgt}$ is larger. Here the slope is 0.51 ‰/$°C$ and $R^2 = 0.668$.

    In this study the $\delta^{18}O_p$-temperature relationship is found insensitive to changes in the pertuba-

tions of sea ice. Interpreting $\delta^{18}O_p$ changes to temperature changes using an Arctic spatial slope will therefore not be dependent on the sea ice distribution in the climate where the spatial slope is estimated. However the results here show that distribution still plays a major role, as the isotopic composition of precipitation at one location might not represent the regional conditions.

### 3.4   Atmospheric moisture processes

The $\delta^{18}O_p$ response to sea ice changes (Fig. 4 ) shows that the response is predominantly local, yet with the "2012" experiment showing a more regional response. We here broadly define a local response as a situation where the grid points in close proximity to regions of sea ice change experience large changes in $\delta^{18}O_p$ and where grid points without sea ice change show no pronounced changes in $\delta^{18}O_p$. Similarly a regional response is here used to describe a response where changes in $\delta^{18}O_p$

both occur at grid points in close proximity to regions of sea ice changes but also at neighbouring grid points without sea ice changes.

    An analysis of future warming in the Arctic using state-of-the-art climate models showed changes in the hydrological cycle due to Arctic warming and sea ice changes (Bintanja and Selten, 2014). Moisture inflow from lower latitudes played a minor role, and the changes was found to be mainly

caused by strongly intensified local surface evaporation. Changes in evaporation of local ocean water have also been suggested by (Sime et al., 2013; Noone, 2004) as important for sea ice induced changes in $\delta^{18}O_p$. Evaporation is important for the control of isotopic composition of the moisture because water evaporated locally in the Arctic Ocean has a different isotopic composition than the surrounding vapour that has been depleted during transport from remote moisture sources.

Examination of the anomalies of isotopic composition of the water vapour yields insight into the isotopic composition of the Arctic moisture. Fig. 7 shows the anomaly of isotopic water vapour composition at the 850 hPa level (here after referred to as $\delta^{18}O_v$). The anomaly is plotted together with the 850-hPa level wind field anomaly overlayed. Similarly to the isotopic composition of precipitation, the isotopic composition of vapour at the 850 hPa level reveals local anomalies at the same

locations as anomalies of sea ice occur for all experiments. Locations with decreased (increased) sea ice extent and concentration match with locations of enriched (depleted) water vapour.





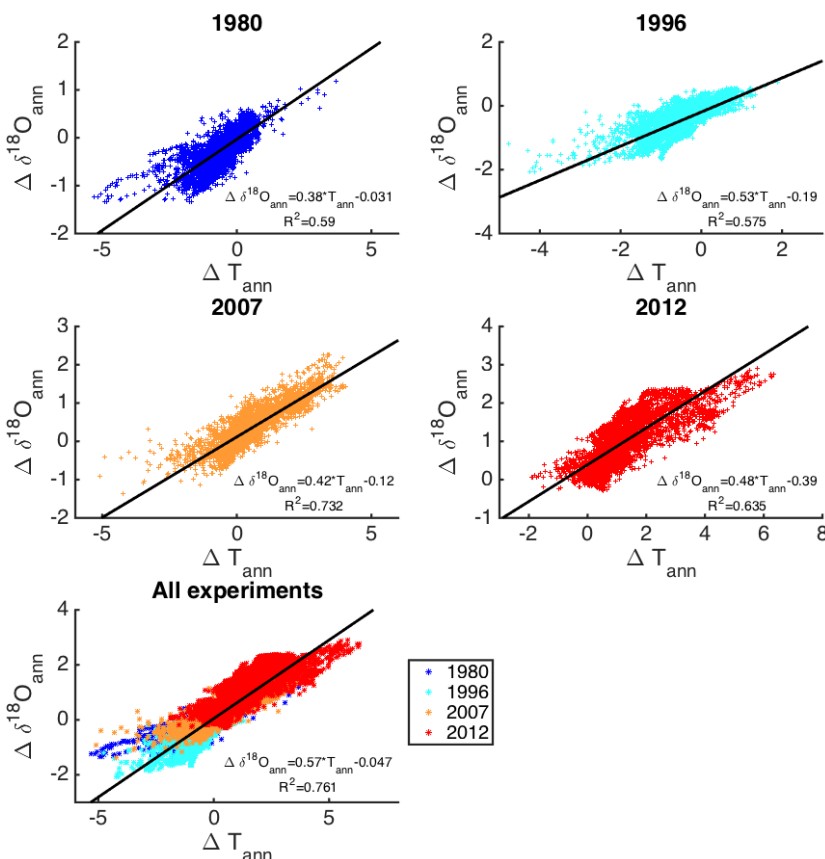

**Figure 5.** Scatter plots of anomalies of annual mean surface temperature ($\Delta T$) versus $\Delta \delta^{18} O_p$ anomalies of all grid points from $60°N$ - $90°N$ for the different experiments compared to CTRL. The colours refer to the different experiments. The regression lines for the individual experiments are shown with colours matching the colours of the markers. Dark blue refers to experiment "1980", light blue to experiment "1996", orange to experiment "2007" and red to experiment "2012".




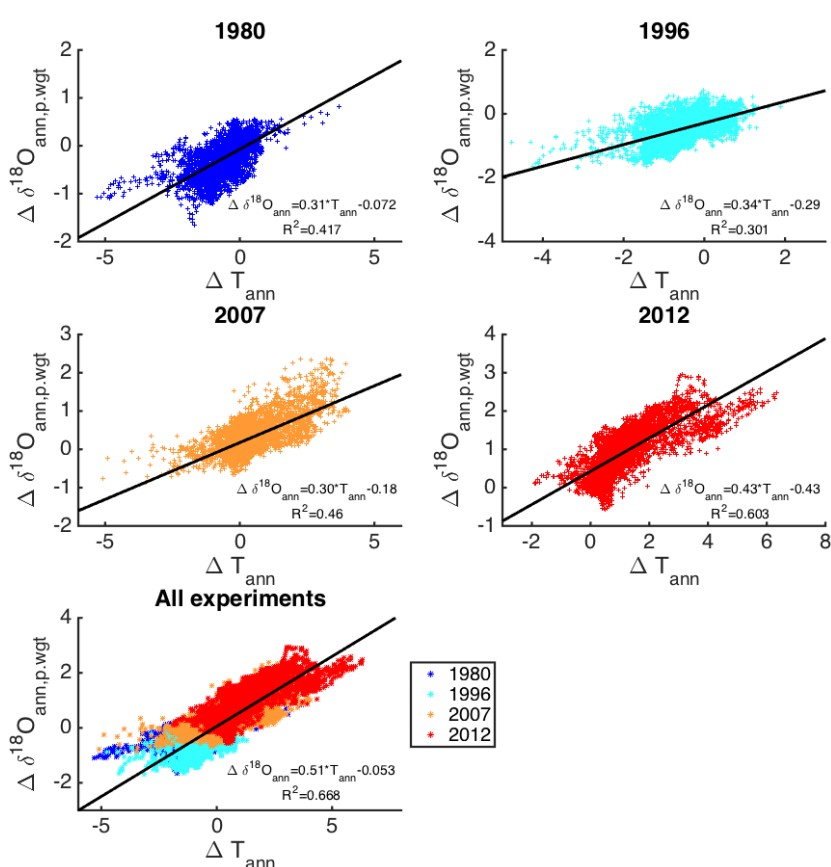

**Figure 6.** Same as figure 5 but for annual mean temperature ($\Delta T$) versus precipitation weighted $\delta^{18}O_p$ anomalies ($\Delta \delta^{18}O_{p,wgt.}$)



This suggests that an increase in local Arctic Ocean evaporation contributes with heavily enriched water to the ambient vapour resulting in vapour that has a higher value of $\delta^{18}O_v$. In the case of an increase in sea ice cover, the decrease in the contribution of local enriched water would result

in ambient vapour with a lower value of $\delta^{18}O_v$. The wind vectors in Fig. 7 show that the changes in advection at the 850 hPa level can not explain the change $\delta^{18}O_v$. Interestingly the highest wind anomalies are found in the "2012" experiment which is also the experiment which displayed a more widespread and regional isotopic response to sea ice changes. The slight increase in local wind anomalies here could indicate that advection are responsible for larger spatial extent of the isotopic

response.

Changes in local evaporation are here investigated using latent heat flux as proxy for the local evaporation. To compare how changes in sea ice cover change the amount of total local evaporation, only locations with grid points of strongly reduced sea ice (change bigger than 20 %) were selected and the amount of total latent heat flux per year for all grid points between $60°N$ - $90°N$ was

calculated for all experiments. To account for different numbers of grid points with sea ice change for each experiment the comparison to the CTRL run is done using identical locations of the grid points, such that non-local effects in evaporation changes were excluded. As observed in Fig. 8 the amount of local evaporation is remarkably stronger for grid points where sea ice is reduced and weaker where sea ice is increased. Thus it is argued that the changes in $\delta^{18}O_p$ over regions of sea

ice change is strongly connected to changes in local evaporation.

The two experiments with low sea ice extent ("2007" and "2012") have warmer temperatures, more intense evaporation and higher values of $\delta^{18}O_v$ than the CTRL experiment. This is in contrast to the two remaining experiments ("1980" and "1996") which have sea ice extent larger than in the CTRL run (based on 1979-2012 mean). In these experiments lower temperatures are observed as

well as less intense evaporation and lower values of $\delta^{18}O_v$ than in the CTRL experiment. Our results confirm that sea ice concentration and SST control the ability of the ocean to evaporate ocean water to the atmosphere.

## 4 Discussion

This analysis shows that changes in sea ice extent yields changes in the contribution of vapour orig-

inating from local moisture sources impacting $\delta^{18}O_v$. Based on pronounced local isotopic response, and the evidence of an increase in evaporation when sea ice is reduced (see Fig. 8), we propose that the primary mechanism behind the pertubed isotopic composition is changes in the contribution from local moisture sources. Since no statistically significant changes are observed in the amount of precipitation it is assumed that the local moisture mixes with existing vapour. This is a local

effect, primarily driven by local sea ice changes and not changes in the remote moisture sources. However, as observed in the "2012" experiment, larger changes in sea ice and more generally Arctic





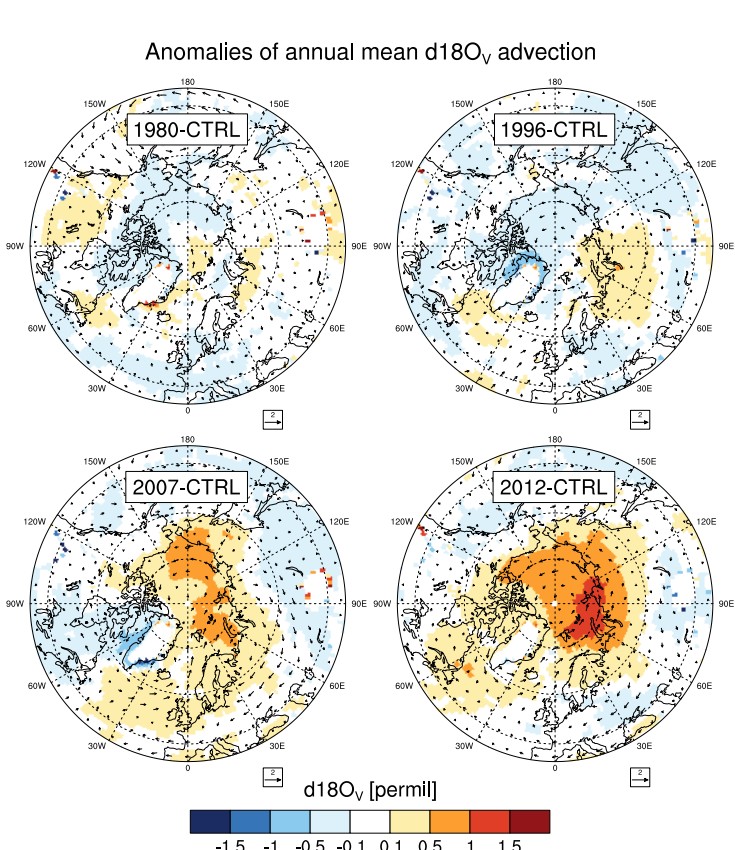

**Figure 7.** $\delta^{18}O_v$ for the four simulations compared to the CTRL run. The arrows show the wind anomalies between the experiments and the CTRL run at the 850 hPa level.





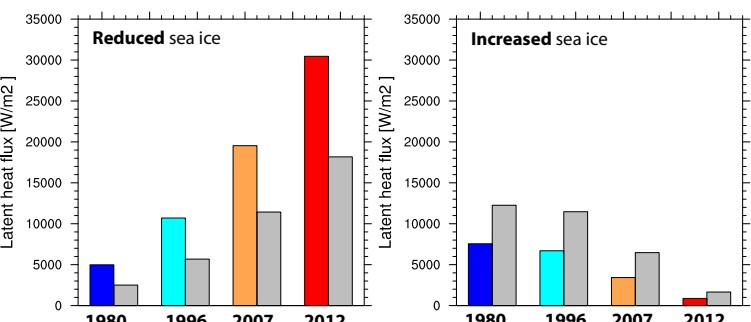

**Figure 8.** Grid points of strongly reduced sea ice (change bigger than 20 %) in each experiment were compared to identical grid points in the CTRL run and the amount of total latent heat flux per year for all grid points between $60°N$ - $90°N$ was calculated for both experiments and the CTRL run. The same was done for grid points of strongly increased sea ice. The coloured bars represent the latent heat flux over sea ice change regions for the different experiments and the grey bars adjacent to the coloured bar represent the latent heat flux for the identical grid points in the CTRL run.

amplification might also induce changes in large scale circulation, which can potentially influence the isotopic signature of remote sources. An illustration of the connection between sea ice, SST and isotopic composition is shown in fig 9.

Existing studies suggest that sea ice impacts - and interacts with - many processes in the atmosphere, affecting cloud cover (Schweiger et al., 2008), the vertical structure of the atmosphere (Graversen et al., 2008), atmosheric water vapour content, the height of the boundary layer, atmospheric convection (Abbot and Tziperman, 2008), the poleward energy moisture transport, the latent heat flux and the connected items: the jet stream, NAO and storm tracks (Singarayer et al., 2006). Based 240    on the findings of Bintanja and Selten (2014) we argue that large scale changes causing enhanced moisture inflow from lower latitudes are neglible here.

The isotopic composition of precipitation in Greenland is sensitive to changes in atmospheric circulation(Vinther et al., 2010; Ortega et al., 2014) and the position of the atmospheric polar front (associated with the North Atlantic jet stream)(Johnsen et al., 1989). In this study it is found that the





position and intensity of the jet stream ( computed as $\sqrt{u^2 + v^2}$ at 300 hPa level) (not shown) varies
in strength and position for the different experiments. A weakening of the midlatitude wind speed
is seen, yet no clear connection to the changes in sea ice extent is found. However this calculation
of annual mean position based on monthly mean output might not be representative of the actual
processes occurring on shorter time scales. This is because sea ice changes have been suggested to

lead to a more meandering jet stream (Liu et al., 2012a; Francis and Vavrus, 2012), altering storm
tracks, storm frequency, regions of cyclogenesis (Vihma, 2014; Bader et al., 2011) and moisture
source region, thus strongly influencing conditions during uptake and transport of moisture. All of
these mechanisms have the potential to alter the isotopic composition of the precipitation.

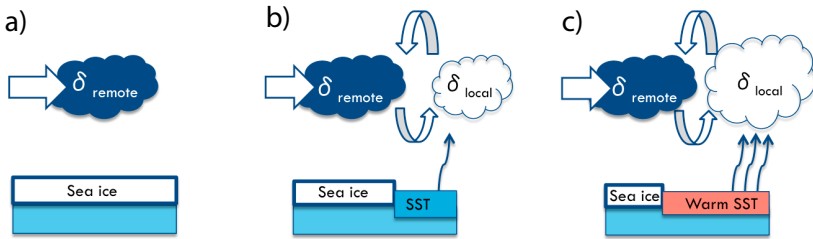

**Figure 9.** This figure display the processes suggested to be the cause of the sea ice influence on the isotopic
composition of Arctic moisture. The three cases a,b and c refer to three different sea ice concentrations: High,
medium and low. The terms $\delta_{remote}$ and $\delta_{local}$ are used to describe the isotopic signature of remote and local
moisture sources. a) In situations with high sea ice concentration all of the moisture in the lower atmosphere
comes from remote sources a the remote isotopic signature is described as $\delta_{remote}$. b) In situations with medium
sea ice concentration evaporation is occurring at the open ocean water region. This water vapour has isotopic
signatures here described as $\delta_{local}$. Also moisture from remote sources is present. Mixing occurs between the
two air masses which also influences the isotopic composition. c) Situations with low sea ice concentration and
warmer SST are somehow similar to b) however as the ocean is warmer the evaporation is more pronounced.
Hence the relative contribution from the two moisture sources with the isotope signatures $\delta_{remote}$ and $\delta_{local}$
are changed. This also yields a more regional extent of the response.

     Changes in isotopes in Greenland precipitation are of special interest due to the deep ice core

research sites in this region. Interestingly, none of the sea ice perturbation experiments in this study
display $\delta^{18}O_p$ changes over central Greenland. We argue that the robustness of the central Greenland
$\delta^{18}O_p$ to changes in Arctic Ocean conditions is related to the topography of Greenland. Specifically,
the steep slopes of the ice sheet margin are associated with substantial orographic enhancement of
precipitation and depletion of storm water vapour content. Intense storms therefore make a dominant

contribution to the water isotopes at the summit of the Greenland Ice Sheet.

     If the Arctic Ocean conditions only form "weak" precipitation systems then moisture transport to
the central part of the Ice Sheet will be reduced. Thus weak locally produced storm systems will
not influence $\delta^{18}O_p$ over Greenland. The Greenland katabatic wind blocking effect (Noel et al.,



2014) might also explain this. Furthermore, no notable changes in sea ice cover are occurring in near proximity to Greenland in this study. This attenuates the contribution of local moisture sources to the Greenland precipitation. However changes in storms tracks and similar features might also influence the location of source region and the isotopic fingerprint of the moisture. (Sodemann et al., 2008b, a; Ortega et al., 2014)

In contrast to the results of this study, Sime et al. (2013) simulated $2-3‰$ changes in central Greenland $\delta^{18}O_p$. This was conducted using SST and sea ice conditions created from coupled model experiment forced by large increases in $CO_2$. The main differences between the simulations in this study and in the study by Sime et al. (2013) is related to the distribution and magnitude of sea ice and SST changes.

In the experiment by Sime et al. (2013) sea ice and SST changes also occur in the region north of Greenland. Also the magnitude of Arctic SST anomalies are $8-10°C$ whereas our simulations have anomalies of $3-5°C$. These differences are compelling as our experiment "2012" with the largest prescribed SST anomalies and sea ice changes also is the only experiment that simulates a regional isotopic response. This indicates that the magnitude of SST changes might control not only the amount of local evaporation, but also the regional extent of the isotopic response. Hence it is possible that the simulated changes of $\delta^{18}O_p$ by (Sime et al., 2013) have a regional extent due to the same reasons as experiment "2012".

Warming of the lower troposphere and associated weakening of inversion layer might be important in controlling the extent of the isotopic response. As sea ice removal is connected to intense warming of the lower troposphere Screen et al. (2012); Deser et al. (2010), it could be speculated that this warming and associated weakening of the inversion layer is controlling the extent of the isotopic response. This would be possible as a weaker inversion layer allows atmospheric convection and Abbot and Tziperman (2008) have shown that this can occur at high-latitudes in sea ice free regions in winter. Further investigation of the mechanism causing this change requires further idealized experiments following a similar to design to Noone (2004) so that a systematic investigation of the atmospheric processes influencing the isotopic composition of moisture is possible.

## 5   Conclusions

The aim of this study was to investigate whether changes in sea ice cover derived from observed anomalies can influence the isotopic composition of precipitation in the Arctic. We have presented results from isoCAM3 an isotope-equipped AGCM, forced with different distributions of Arctic sea ice changes and associated SST from the ERA-interim re-analysis product. These simulations clearly show that a changed sea ice cover influences the isotopic composition of Arctic precipitation with regional changes of $\delta^{18}O_p$ of up to $3‰$ in the Barents Sea region. For all experiments we find that



increased (decreased) sea ice extent and concentration results in enriched (depleted) $\delta^{18}O$ values of precipitation.

Our simulations show that in isoCAM3 the $\delta^{18}O$ response to the ocean conditions is primarily local. Within the same experiment large changes in $\delta^{18}O$ are observed over some regions and no changes over other regions. The geographical variations in the $\delta^{18}O$ response to Arctic sea ice changes show that the isotopic composition of Arctic precipitation is sensitive to the spatial distribution of the sea ice and SST changes. This means that different distributions of similar sea ice areas can produce very different $\delta^{18}O_p$ values at the same location. Or conversely, that different locations respond very differently in $\delta^{18}O_p$ to the same total Arctic sea ice extent.

For example the isotopic composition in precipitation for the central part of Greenland appears unaffected by the imposed changes in sea ice cover in all experiments. As many ice cores originate from the Greenland Ice Sheet this is an important result for the interpretation of isotope records. Our analysis indicates that the changes in $\delta^{18}O_p$ and $\delta^{18}O_v$ could be (partly) explained by changes in local moisture sources. As the interior Greenland Ice Sheet has a continental climate with no local moisture sources it is suggested that this could explain why the $\delta^{18}O_p$ in central Greenland is relatively insensitive to the distribution of the sea ice changes used in this model study. The spatial $\Delta\delta^{18}O_p$ - $\Delta T$ relationship was found to be unaffected by the distribution and magnitude of the sea ice changes in this study.

Previous studies have shown that large changes in the state of sea ice and SST conditions influences the isotope composition over Greenland (Sime et al., 2013) and Antarctica (Noone, 2004) but this study is the first model experiment to show that minor (relative to Sime et al. (2013)) perturbations in the sea ice cover and SST under present-day climate conditions can yield significant changes in the isotopic composition of precipitation in the Arctic.

*Acknowledgements.* The research leading to these results has received funding from the European Research Council under the European Union's Seventh Framework Programme (FP7/2007-2013) / ERC grant agreement number 610055 as part of the ice2ice project. The authors acknowledge the support of the Danish National Research Foundation through the Centre for Ice and Climate, Niels Bohr Institute.



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
