# Peer review of "How does sea ice influence $\delta^{18}\text{O}$ of Arctic precipitation?"

_Atmospheric Chemistry and Physics, 2016_

## Referee Comment (RC1) · Anonymous Referee #1 · 22 Mar 2016

**Comments on "How does sea ice influence $\delta^{18}O$ of Arctic precipitation? " by Faber et al**

March 22, 2016

This paper investigates the joint influence of sea ice cover and sea surface temperature on precipitation $\delta^{18}O$ in the Arctic, based on 5 simulations with an isotopic general circulation model. The idea is very interesting but I find the analysis is shallow and disappointing. I recommend publication after major revisions. Below are some suggestions to improve the paper.

**1 Major comments**

**1.1 The abstract is misleading**

- The abstract emphazing the influence of sea ice on $\delta^{18}O$ of Arctic precipitation except on central Greenland is misleading. Actually, there is almost no change in $\delta^{18}O$ over most of Greenland, even coastal Greenland. The main conclusion of this paper, which deserves to be emphasized in the abstract, is that moderate changes in Arctic sea ice conditions will have no effect on $\delta^{18}O$ over most of Greenland, especially where ice cores are available.

- Same in the conclusion. e.g. l 317-320: "significant changes", but it should be emphasized that they are mainly local, not where most of the ice cores are.

- l 313: "relatively" -> "completely"?

**1.2 Need for model evaluation in the Arctic**

- Before using a model, some evaluation of this model is necessary. Can you add a figure showing the distribution of precipitation $\delta^{18}O$ in the Arctic compared to precipitation data wherever they are available? (GNIP, snow samples, ice cores...)

- If the model features some biases, what are the consequences on the conclusions? For example, is the absence of large-scale isotopic response specific to this model, and can it be linked to the representation of the large-scale circulation or of boundary layer processes?

**1.3 Sea surface conditions rather than sea ice cover**

- The authors discuss the sensitivity to sea ice conditions, but actually they cannot separate the effects of sea ice cover and sea surface temperature. I suggest they reword all sentences by replacing sea ice by sea surface conditions. e.g. l 134 and elsewhere.

**1.4 Question on precipitation weighting**

- l 156: The precipitation weighting is not clear. Isn't $\delta^{18}O_p$ weighted by precipitation already on Figures 4 and 5? If not, what does it represent? Precipitation $\delta^{18}O_p$ should always be precipitation weighted. Otherwise, biases are introduced that depend on the arbitrary choice of the temporal sampling. If you didn't weight by precipitation, this is a mistake. Then once $\delta^{18}O_p$ is precipitation weighted, you don't need to further apply any precipitation weighting, so fig 6 is useless.

**1.5 $\delta^{18}O_p$-temperature relationships**

- l 161: the scatter diagrams in fig 5 are interesting but underexploited. The authors calculate the slope for all simulations together but don't do anything with it. What is interesting in this diagram is that in spite of the similar slope, the different simulations appear to be shifted. For a given surface air temperature, the $\delta^{18}O_p$ is all the larger as the sea ice is overall reduced. This explains why the slope for all simulations together is larger than that for individual simulations. There is an offset that depends on sea surface conditions and that adds up to the temperature effect. This is interesting.

- As a consequence, I disagree with the implications for the interpretation of $\delta^{18}O_p$ signals written by the authors l 169-174.

  - "In this study the $\delta^{18}O_p$-temperature relationship is found insensitive to changes in the pertubations of sea ice": the slope is the same but there is an offset, which is very important for the interpretation of $\delta^{18}O_p$ temporal signals!

  - "Interpreting $\delta^{18}O_p$ changes to temperature changes using an Arctic spatial slope will therefore not be dependent on the sea ice distribution in the climate where the spatial slope is estimated". Do you mean interpreting temporal changes at a given location? If so, this assertion is wrong. Because of the offset, the temporal changes in $\delta^{18}O_p$ at one location will show a larger slope with temperature than expected from the Arctic spatial slope, whatever the climate used for the slope estimation. In other words, the effect of sea ice is to disturb the $\delta^{18}O_p$-temperature relationship so that the temporal relationships will look stronger than the spatial relationships.

- l 314: revise this conclusion.

**1.6 The link with vapor origin is not clear**

- l 200-204: to test the hypothesis of an advective effect, you can analyze wether the $\delta^{18}O_v$ signals are downstream or upstream compared to the wind directions.

- The discussion on air mass origin is not clear. l 224-225: "changes in the contribution of vapor origin": if it was the case, shouldn't we expect an isotopic response at a more regional scale? l 224: "shows": how can you eliminate the hypothesis that all the $\delta^{18}O_p$ response is due to temperature?

-> The message about the influence or about the absence of influence of a change in air mass origin should be clarified and better supported by model analysis.

- Symptomatic of this unclear message: l 233: the authors mention "large scale circulation". Then they refer to Fig 9, where there large-scale circulation is not discussed.

- l 235-241: what is the link between these references and this study? How do these references support your point or advance the discussion?

- l 253: "all these mechanisms have the potential to alter the isotopic composition." But do they play a role in this study?

- l 255-268: This is not clear. What is the link between storm systems and moisture sources? Can you support it by model analysis?

-> It would be useful to make a clear distinction between what your model analysis actually shows, with rigorous analysis, and what you can speculate based on the litterature.

- l 310-311: it looks like moisture sources don't explain much of the isotopic signal based on the model analysis. Comparing Fig 3 and Fig 4, it looks like $\delta^{18}O_p$ simply follows air temperature. So why do you need to involve the effect of moisture source?

**1.7 Some suggested additional analysis to clarify the link with large-scale circulation**

- One hypothesis to explain the absence of isotopic response in Greenland is that air trajectories that end up in Greenland travel in the free troposphere above the boundary layer, so are not sensitive to sea surface conditions and are not recharged by Arctic evaporation. Can you test this hypothesis in the model? For example, you can look at isotopes and large-scale circulation for vertical cross-sections from the ocean to central Greenland?

- l 282-290: this is interesting. What is the vertical distribution of $\delta^{18}O_v$ response in your model? This is easy to look at in your simulations. Looking at this will help you to understand why the isotopic response is so local.

**2 Miscelaneaous**

- l 18: add more references, including the key historical ones

- l 32: "demonstrate" -> "suggest": Observations don't demonstrate anything in absence of some form of modelling

- l 64: The citations for the isotopic versions of CAM3 are wrong: [1] was for MUGCM; Noone 2003 was a workshop with no written record; [2] only cites Noone 2003. Same problem l70.

- l 67: what is a 3rd generation isotope scheme? This is not a commonly accepted classification. Be more specific or don't mention it.

- l 130-132: "match the spatial distribution": to be more quantitative, calculate the spatial correlation between SST and $\delta^{18}O$ changes. l 133-l152 don't add much to the discussion once we have read the first paragraph and looked at the figures. The correlation would add a quantification.

- l 156: "precipitation weighted $\delta^{18}O_{p.wgt}$" -> "precipitation weighted $\delta^{18}O_p$, named $\delta^{18}O_{p.wgt}$,"

- Figure 3 caption: are these annual mean?

- Figure 4 caption: are these annual mean?

- l 205: Latent heat flux is not a proxy for evaporation, they are actually the same, ignoring a multiplication factor.

- Figure 8: how many grid points are considered for each year? Can you add this information? Can you add error bars? (standard deviation)

- l 246: "weakening": for which years? Aren't there any years with strengthening of the wind speed? If so, there could be an effect of looking at individual years versus using climatological SSTs.

**References**

[1] D. Noone and I. Simmonds. Associations between delta18O of Water and Climate Parameters in a Simulation of Atmospheric Circulation for 1979-95. *J. Climate*, 15:3150–3169, November 2002.

[2] David Noone and Christophe Sturm. *Comprehensive Dynamical Models of Global and Regional Water Isotope Distributions, in Isoscapes: Understanding movement, pattern, and process on Earth through isotope mapping.* Springer Netherlands, 2010.

---

## Referee Comment (RC2) · Anonymous Referee #2 · 6 May 2016

In this manuscript, the authors investigated the impact of sea ice and SST distributions in an isotope-incorporated general circulation model (IsoCAM3). The results indicated that less sea ice leads more enriched d18O in precipitation, but mostly over the areas where sea ice changed. Inland areas, such as central part of Greenland, there was almost no change in d18O in P. That was a unique finding of them because previous studies showed that at central Greenland d18Op changed by surface temperature.

However, in my opinion, it was premature to conclude there was almost local impact only on d18Op in Greenland. The experiments they conducted were using the same SST and sea ice distribution over non-Arctic regions. The atmospheric fields of four experiments could be essentially very similar each other over not only non-Arctic but also Arctic. If so, the impact of sea ice could be only local because the general circulation was constrained. In other words, sea ice had little impact to the large scale moisture

transport.

Therefore, this reviewer would like to request the authors to conduct additional experiments using globally different SST and sea ice distribution in addition to the current experiments. By doing so, it can be concluded that whether d18Op over Greenland cannot be influenced by SST and sea ice over the region. I believe that this is a main reason of the difference from previous studies (i.e., d18Op change over Greenland was insignificantly related with sea ice change). That is my major request.

There are relatively smaller requests, too. 1. L67: What is third-generation isotope scheme?

2. L82: How did the initial condition prepared?

3. L90: As written above, Arcitic oceaninc surface boundary conditions may not so significantly influence the general circulation. Please check.

4. L115: I could not understand, "this would also smooth out naturally occurring SST gradients".

5. L127: In addition to Figure 3, please show anomalies of precipitation.

6. L201: It is hard to see the anomalies in wind speed from Figure 7.

7. L255: From the experiment, there was no impact in d18Op over central Greenland. However how about the reals situations? There is no temperature change, too? Please check.

8. L314: How about temporal tendency in Delta-d18O and DeltT? How about in reality? Please check.

9. L319: What's the major difference in this model compared to Sime et al. (2013)?

---

## Author Comment (AC1) · 27 Jun 2016

The comment was uploaded in the form of a supplement:
http://www.atmos-chem-phys-discuss.net/acp-2016-100/acp-2016-100-AC1-supplement.zip

---

## Author Comment (AC2) · 27 Jun 2016

The comment was uploaded in the form of a supplement:
http://www.atmos-chem-phys-discuss.net/acp-2016-100/acp-2016-100-AC2-supplement.zip

---

## Author Response (AR1)

**Authors' response to comments on paper: How does sea ice influence $\delta^{18}O$ of Arctic precipitation?**

Anne-Katrine Faber et al.

We thank the referees and the editor for their comments made to our manuscript. We appreciate the constructive feedback on our manuscript and have made substantial changes to the manuscript.

**Response to anonymous referee # 1:**

**1.1 Abstract misleading - Focus on Greenland**

**Question:**

The abstract emphazing the influence of sea ice on  $\delta$  180 of Arctic precipitation except on central Greenland is misleading

**Changes in manuscript:**

Changes in sea ice and sea surface temperatures have different impact in Greenland and the rest of the Arctic. The simulated changes in central Arctic sea ice does not influence \$\delta^{18} O\$ of Greenland precipitation, only anomalies of Baffin Bay sea ice. However, this does not exclude that simulations based on other sea ice and sea surface temperature distributions might yield changes in Greenland \$\delta^{18} O\$ of precipitation.

**Question:**

Same in the conclusion. e.g. I 317-320: "significant changes", but it should be emphasized that they are mainly local, not where most of the ice cores are.

**Changes in manuscript:**

**Conclusion:**

The geographical variations in the \$\delta ^{18} O\$ response to changes in Arctic sea surface conditions show that the isotopic composition of Arctic precipitation is sensitive to the spatial distribution of the sea ice and SST changes, however not at Greenland

The isotopic composition of Greenland precipitation are unaffected by the imposed changes in central Arctic sea ice cover in all experiments. Only conditions near Baffin Bay influence Greenland.

**Question:**

L313: relatively -> completely? *Changed.*

**1.2 Need for model evaluation in the Arctic.**

**Question:**

**Before using a model, some evaluation of this model is necessary. Can you add a figure showing the distribution of precipitation $\delta$ 180 in the Artic compared to precipitation data wherever they are available? (GNIP, snow samples, ice cores...)**

**Changes in manuscript:**

A figure is added where data from ice cores and coastal Greenland GNIP stations are compared to the CTRL run

**Question:**

If the model features some biases what are the consequences on the conclusions. For example, is the absence of large-scale isotopic response specific to this model, and can it be linked to the representation of the large-sale circulation or of boundary layer processes?

**Comments:**

The model is positive biases producing to enriched d18O over Greenland as many other models.

To answer whether the absence of a large-scale isotopic response is specific to this model would require model-intercomparison study. Currently no model-intercomparison studies of isotope-enabled GCM are published for Arctic conditions.

**1.3 Sea surface conditions rather than sea ice cover Request:**

Use the term sea surface conditions rather than sea ice cover

**Changes to manuscript**

This is changed through out the paper. Either sea ice cover and sea surface temperatures are used together, or the term sea surface conditions is used the describe both.

**1.4 Questions on precipitation weighting.**

**Changes to manuscript**

All d18O values from model output are shown as precipitation weighted. This is now corrected in the manuscript.

**1.5 d18O-Temperature relationships.**

**Comments:**

Reviewer 1 discusses the importance in the differences in the intercept values for the different experiments. In the previous version of figure 5, the plot "All experiments" where each experiment were plotted with different colors. Therefore due to the plotting routine the differences in intercepts for each experiment incorrectly looked more pronounced. Therefore the plot "All experiments" in fig. 5 now plots all values with the same color. However, the differences in the intercepts values, especially for experiments "2012" is now added to the manuscript

The reviewer disagrees with the implications for the interpretation of the d18O p signal. This part is now only briefly mentioned in the paper, and the manuscript now only mentions that the slope of  $delta ^{18} O_{p}$ -temperature relationship is found to be insensitive to changes in the perturbations of sea ice.

No further analysis on this topic is conducted as the focus on this manuscript is directed towards a discussion on the causes of changes in the isotopic response.

**Changes in the manuscript**

**1.6** The link with vapor origin is not clear**

**& 1.7 Clarify the link with large-scale circulation Comments:**

We clearly agree with the reviewer on this comment. Further analysis has been made and the results and discussion sections have been rewritten in order to improve this link.

**Changes in the manuscript**

To clarify the influence of the observed simulated change in d18O and the connection to either change in air mass origin or local temperature then analysis of the vertical distribution of T and d18O have been made. The zonal cross sections at latitude band 77 N have been added to the manuscript (fig 8 and fig 9). The given latitude has been selected to match the nearest grid point to the location of the ice core drilling site NEEM at Greenland. NEEM is selected rather than central Greenland due to several reasons. First, because the latitude band 77N covers a circumpolar band with regions of large sea ice changes all over the Arctic. Second, recent observations from Steen-Larsen 2011 find a connection between the isotopic signal at NEEM and Baffin Bay sea ice extent.

Furthermore spatial fields of d18Ov are added to the appendix. These show d18Ov at two different pressure levels, 950hPa and 700 hPa (thus representing different layers in the vertical) and show that clear surface based signal is found all over the Arctic and not just at the given selected 77 N latitude band as shown in the cross section plot.

In short we find that anomalies of d180v are surface based and connected to grid points of changes in sea ice. But changes are also seen for temperature near the surface. We cannot separate the effect of temperature and changes in moisture source in this study due to the lack of moisture tracking. Therefore the discussion in this paper is now treating the possibilities of either changes in moisture source or temperature – but no conclusion is made.

Structural changes have been made to the manuscript in order to separate the findings from this set of model experiments and speculations based on findings from other studies.

**2. Miscellaneous.**

**L18: Add more references, including key historical ones Changes in the manuscript**

Since the pioneering work by Dansgaard (1964), the understanding of stable water isotopes as a proxy for temperature has significantly advanced. It has become clear that the isotopic composition of precipitation is a complex signal, influenced by both local and regional climate conditions (Vinther et al., 2010; Steen-Larsen et al., 2011; Sjolte et al., 2011; Sodemann et al., 2008b; White et al., 1997; Johnsen et al., 1989)

**L32: Demonstrate -> suggest OK L64: Citations are wrong for isoCAM3:**

**Comments:**

A model release paper does not exist for isoCAM3 thus it chosen to refer to Noone and Sturm 2010 as also done by other studies.

**Changes in the manuscript**

More details of isoCAM3 can be found in Noone and Sturm (2010)

**L67: What is third generation isotope scheme?**

This is now removed as this is not relevant

**L156 – precipitation weighted d18Opwgt, Corrected Fig 3 and figure 4 caption – are these annual means? Yes, corrected**

L205 – latent heat flux as a proxy for evaporation

**Changes in the manuscript**

Changes in local evaporation are here investigated based on the surface latent heat flux

**Figure 8 (now figure 10).**

Additional statistical information required for this analysis. Changes in the manuscript

The number of grid points of reduced sea ice is as follows; 1980: 217, 1996: 444, 2007:1148, and 2012: 2116. And the number of grid points of increased sea ice; 1980: 1508, 1996: 1024, 2007:554, 2012: 437.

**L246 Weakening of the jet stream for separate years...**

**Comments:**

Investigating the differences in the variability within the weakening of the Jetstream could yield information on the control of the sea surface conditions on jet stream variability. However given the model setup in this experiments with artificially constructed ocean data sets consisting of an Arctic section and a non-semi Arctic mean values section (as described in L109-L116) it is found not favorable to focus on Jetstream conditions as it is uncertain whether the potential artificially introduced SST gradients near 37 N in the Atlantic might alter the representation of the jetstream

Manuscript prepared for Atmos. Chem. Phys. with version 2015/09/17 7.94 Copernicus papers of the LATEX class copernicus.cls. Date: 29 June 2016

**How does sea ice influence $\delta^{18}O$ of Arctic precipitation?**

Anne-Katrine Faber1, Bo Møllesøe Vinther1, Jesper Sjolte2, and Rasmus Anker Pedersen1,3

1Centre for Ice and Climate, Niels Bohr Institute, University of Copenhagen, Copenhagen, Denmark

2Department of Geology, Quaternary Sciences, Lund University, Lund, Sweden 3Climate and Arctic Research, Danish Meteorological Institute, Copenhagen, Denmark

*Correspondence to:* Anne-Katrine Faber (akfaber@nbi.ku.dk)

Abstract. In this study we investigate the influence of variations in sea ice cover on the isotopic composition of precipitation in This study investigates how variations in Arctic sea ice and sea surface conditions influence  $\delta^{18}O$  of present-day Arctic precipitation. This is done using the model isoCAM3, an isotope-equipped version of the National Center for Atmospheric Research Commu-

5 nity Atmosphere Model version 3. Four sensitivity experiments and one control simulation are performed with prescribed SSTs SST and sea ice. Each of the four experiments simulates the atmospheric and isotopic response to Arctic oceanic conditions for selected years after the beginning of the satellite era in 1979.

Results show that the isotopic composition of Arctic precipitation Changes in sea ice and sea

- 10 surface temperatures have different impact in Greenland and the rest of the Arctic. The simulated changes in central Arctic sea ice does not influence  $\delta^{18}O$  of Greenland precipitation, only anomalies of Baffin Bay sea ice. However, this does not exclude that simulations based on other sea ice and sea surface temperature distributions might yield changes in Greenland  $\delta^{18}O$  of precipitation. For the Arctic,  $\delta^{18}O$  of precipitation and vapour is sensitive to changes in sea ice extentlocal changes
- 15 of sea ice and sea surface temperature and the changes in vapour are surface based. Reduced sea ice extent yields more enriched isotope values, while increased sea ice extent yields more depleted isotope values. Results also demonstrate that the configuration The distribution of the sea ice cover anomalies are important and sea surface conditions is found to be essential for the spatial distribution of the isotopic response. The configurations of sea ice change used in these simulations did not show
- 20 a change in simulated changes in  $\delta^{18}O$  for central Greenland. However, results indicate that other configurations of sea ice changes could yield different results for Greenland.

**1 Introduction**

Stable Records of stable water isotopes from polar ice cores have been widely used to reconstruct past climate variability. Since the pioneering work by ?, the understanding of stable water isotopes as

- 25 a proxy for temperature have undergone a significant developmenthas significantly advanced. It has become clear that the isotopic composition of precipitation is a complex signal both influenced by influenced by both local and regional climate conditions (??) (??????). The isotopic composition of the precipitation is an integrated measure of the regional climatic and atmospheric conditions throughout the transport of moisture integrated along the moisture transport pathway from source to
- 30 deposition. Accordingly As a result, there is a need for a detailed process-based understanding of the factors that can alter the isotopic composition of the transported moisture.

The physical and dynamical processes that influence the isotopic composition of precipitation have been investigated Studies using models, ice cores, snow and vapor measurements. Results indicate that variations vapour measurements have investigated the physical and dynamical processes

- 35 influencing the isotopic composition of precipitation. Variations in local Greenland temperatures, conditions at source regions , North Atlantic Oscillation (NAO) and air mass trajectories and atmospheric circulation all influence the isotopic composition of Greenland precipitation (???????). Based on studies of Antarctic and non-Greenland Arctic ice cores, sea ice extent have been qualitatively estimated to be essential for the isotope composition of precipitation (?????).
- 40 Several general circulation model (GCM) studies have stressed the importance of sea ice as essential for understanding the model studies highlight sea ice changes as important for understanding changes in the isotopic composition in precipitationover Greenland both of precipitation. Sea ice changes in the Arctic were investigated during Dansgaard-Oeschger events (?) and for exceptionally warm climates (?). Currently no model studies exist of the isotopic response to the recent observed
- 45 drastie decline in Arctic sea ice or for observations in Antarctica. However in the latter the isotopic response to For Antarctica, the impact of sea ice changes has been investigated by modeling idealized were studied using idealized reductions of the circular shaped sea ice cover (?). None of these model studies investigate sea ice perturbations comparable to present-day observations. Measurements from ice cores spanning this period suggest that sea ice changes (?). Results herein showed that the
- 50 sea ice changes led to changes in diabatic heating of the atmosphere which influenced atmospheric circulation. The large spatial differences in can influence the isotope composition of precipitation (????).

A study of idealised changes of Antarctic sea ice show a non-uniform spatial distribution of the modelled isotopic response over Antarctica was-(?). The heterogeneity of the response is sug-

55 gested to reflect the existence of different processes driving local and long range moisture transport to coastal and high elevation regions of Antarctica. Due to differences in the configuration of landmasses, open ocean and sea ice, it is however difficult to directly transfer findings of ? from Antarctica Antarctic to the Arctic. The study of (?) have investigated the sea ice influence on isotopes in Greenland. It was found

- 60 that different impact of changes in sea ice and connected sea surface temperatures (SST) of the Arctic ocean were studied by ?. The sea ice conditions were created using an experiment where a coupled climate model was forced by respectively  $2\times$ ,  $4\times$  and  $8\times$  CO2. Hereafter the sea ice and SST conditions were used to force the applied atmospheric isotope models. Differences in the configurations of sea ice extent and the corresponding warming of the Arctic ocean was SST were
- 65 found to be essential for the resulting large variability in the isotope-temperature slope of 0.1 0.7 %0/C for the Greenland ice sheet. While the 2×, 4× and 8 × CO2 forced model-setup of ? disallows for these CO2 changes used by ? do not allow direct comparison with present-day Arctic conditions, the results highlights highlight processes that might be important for other elimateconditions too. Hence results suggest that the differences in spatial distribution of the sea ice and ocean warming
- 70 can control the relative influence present day climate.

The recent decades of rapid Arctic sea ice decline provide an interesting opportunity to study how  $\delta^{18}O$  respond to realistic changes of sea ice change at the given location.

The results from ? and ? suggest that there are spatial differences in the response to sea ice changes. Accordingly this has implications for the interpretation of ice cores. Understanding of

- 75 how sea ice influences the isotopic composition can thus potentially improve the isotope-based elimatereconstructions. Using model-based experiments to investigate this topic are an important addition to observational studies because the use of an isotope-enabled AGCM makes it possible to scrutinize the physical processes causing the simulated isotopic response. This has the advantage that such studies yields information on the spatial differences in the response that would not be possible
- 80 with ice core data only. Therefore the isotopic response can be analyzed at all grid points in the model instead of the response at the very few locations where ice cores were drilled.

and sea surface temperatures of present-day climate. We here present results from isotope isoCAM3 model simulations forced with observed Arctic sea ice and sea surface temperature (SST) conditions derived from observations. This paper will address how the sea ice extent is influencing the isotopes

- 85 and sea surface conditions influence the  $\delta^{18}O$  in precipitation in the Arctic, and if the response depends on the the role of the spatial configuration of the sea ice changes. Furthermore the processes causing the changes in the isotopes will be investigated. This is done for several configurations of sea ice to asses the robustness of the response to different magnitudes of sea ice concentration changes and spatial patterns. The focus of the analysis of isotopic response is in this study limited to  $\delta^{18}O$
- 90 onlysurface changes. The structure of the paper is as follows; (1) The model and experiments are described, (2) Results of the simulations are presented, (3) The influence of atmospheric moisture processes is discussed.

**2 The model and experimentsExperimental configuration**

**2.1 The model isoCAMisoCAM3**

95 The simulations of the isotopic composition of precipitation and water vapor vapour in this study are conducted with isoCAM3(???)... This is an atmospheric general circulation model (AGCM) enabled with the ability to trace the various species of water isotopes. The model is based on the Community Atmosphere Model version 3 (CAM3) (?)with a third-generation isotope scheme.

The hydrological cycle in CAM3 has been found to be efficient in simulating the overall structure

- 100 of the hydrological cycle (?). The performance of the isotope-module in , and the isotope module was developed by David Noone, University of Colorado. More details of isoCAM3 has been investigated with a model-data comparison (?). The present-day simulations from this study showed a good agreement on global and regional scales with observed spatial isotopic patterns from the database Global Network of Isotopes in Precipitation (GNIP). Furthermore can be found in ? The model iso-
- 105 CAM3 has been applied in several studies that investigated the isotopic response to past climate changes (??????) (???????).

**2.2 The experiments**

To investigate the influence of changes in the Arctic sea ice cover on stable water isotopes, four sensitivity experiments and one control simulation are performed. The horizontal resolution of the

- 110 model is T85 (~ 1.4° x 1.4°) with 26 hybrid-sigma levels in the vertical. All of the simulations are run for 15 years (excluding one year for spin-up). Every run has identical modern day boundary conditions for orbital configurations, ice sheets and greenhouse gases (GHGs), the latter In this study the SST and sea ice concentrations are specified, thus the only surface temperatures that are calculated interactively are land and sea ice surface temperatures. This configuration allows no
- 115 feedback between atmospheric circulation and open ocean SST. Greenhouse gases, vegetation, ice sheets are all set to modern conditions. More specifically greenhouse gases are set to the following CAM3 default levels (year 1990): CO2: 355 (ppmv), CH4: 1714 (ppbv), N2O: 311 (ppbv). The solar constant is set to the CAM3 default settings corresponding to GHG levels in 1365 (Wm-2) and orbital configurations are set to the year 1990. Only the Arctic oceanic surface boundary conditions
- 120 differ between the runs. 1850.

**2.2 Ensemble design**

We perform a set of four sensitivity experiments and one control simulation to investigate how observed variations in Arctic sea surface conditions influences  $\delta^{18}O$ . Every model integration is run for 15 years (following one year for spin-up). Each of the four runs simulates the atmospheric and

125 isotopic sensitivity experiments simulates the  $\delta^{18}Q$  response to sea ice conditions concentration and sea surface temperature (SST) for selected years in the time period 1979-2013 within the satellite era(1979-2012). The years are chosen as the. The 12-month time periods are selected based on the four most extreme cases of sea ice changes based on satellite data of high and low September sea ice extent (recorded during the time period (1979-2012) by the NSIDC Sea ice Index (?) updated daily.

- 130 The two years with highest September sea ice extent is 1980: 7.8 mio. km2 and 1996: 7.9 mio km2. The two years with lowest (?, updated daily). The control simulation (CTRL) simulates the  $\delta^{18}O$ response using the 12 months climatology of sea ice concentration and SST for the full time period April 1979 to March 2013. Only the Arctic oceanic surface boundary conditions differ between the runs. An overview of the model experiments are given in table ??.
- 135 We force the model isoCAM3 with an annual cycle of monthly mean SST and sea ice conditions obtained from ERA-Interim (?). This annual cycle goes from April to March thus spanning the full sea ice cycle related to the selected cases of September sea ice extentis 2007: 4.3 mio. km2 and 2012: 3.6 mio. km2. Here after the model runs for 15 years (following one year of spin up) with repeated annual cycle. The re-analysis data are interpolated bilinearly from the ERA-Interim (1° x 1°) to the
- 140 CAM3 T85 resolution, and hereafter checked for consistency.

Changes in SSTs in the Arctic region-

**Overview of model experiments**

| Experiment      | Prescribed SST and sea ice                                  |
|-----------------|-------------------------------------------------------------|
| ''1980'' | ERA-Interim monthly mean: April 1980-March 1981             |
| ''1996'' | ERA-Interim monthly mean: April 1996-March 1997             |
| ''2007'' | ERA-Interim monthly mean: April 2007-March 2008             |
| "2012"   | ERA-Interim monthly mean: April 2012-March 2013             |
| CTRL            | ERA-Interim monthly mean climatology: April 1979-March 2013 |

Table 1. Overview of model experiments

Changes in Arctic SST are in nature an inseparable part of the sea ice changes. Keeping the SSTs SST constant and only simulating the atmospheric response to sea ice changes, would therefore lead to unrealistic temperature gradients See (see ? for further discussion on this topic. Therefore).

- 145 Therefore, we chose that these experiments are based on both changes in sea ice and SSTs. Data are obtained from ERA-Interim (?) To SST. A masking of the SST data is applied to eliminate remote influences from extra-polar climate patterns (e.g. from the El Niño Southern Oscillation or Pacific Decadal Oscillation)the model is forced with prescribed SSTs and sea ice. This masking is constructed so that only the conditions near the Arctic differ from experiment to experiment.
- 150 Hencethis global data ocean data set, this global ocean data is divided in an Arctic and a non-Arctic region. The Arctic region refers to the region of ocean/sea ice conditions expected to influence the Arctic climate and is therefore rather semi-Arctic. Due the geographical configuration of the